# CompeteAI: Understanding the Competition Dynamics of Large Language Model-based Agents

**Qinlin Zhao** [1]  **Jindong Wang** [2]  **Yixuan Zhang** [3]  **Yiqiao Jin** [4]  **Kaijie Zhu** [2]  **Hao Chen** [5]  **Xing Xie** [2]

## Abstract

Large language models (LLMs) have been widely used as agents to complete different tasks, such as personal assistance or event planning. While most of the work has focused on cooperation and collaboration between agents, little work explores *competition*, another important mechanism that promotes the development of society and economy. In this paper, we seek to examine the competition dynamics in LLM-based agents. We first propose a general framework for studying the competition between agents. Then, we implement a practical competitive environment using GPT-4 to simulate a virtual town with two types of agents, including restaurant agents and customer agents. Specifically, the restaurant agents compete with each other to attract more customers, where competition encourages them to transform, such as cultivating new operating strategies. Simulation experiments reveal several interesting findings at the micro and macro levels, which align well with existing market and sociological theories. We hope that the framework and environment can be a promising testbed to study the competition that fosters understanding of society. Code is available at: https://github.com/microsoft/competeai.

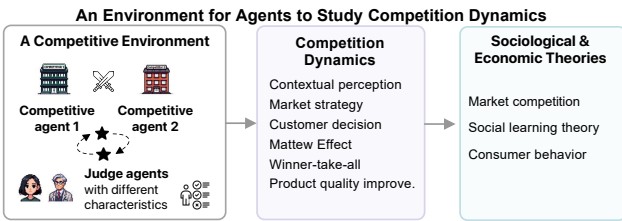

*Figure 1.* Our environment studies competitive dynamics, aligning with established sociological and economic theories.

## 1. Introduction

Competition is a key driving force shaping human societies and influences various domains such as economics, social structures, and technology development. Understanding these competition mechanisms is essential to understand how societies function. Traditional research to study competition has been based mainly on empirical studies (Phan et al., 2019; Markussen et al., 2014). Constrained by the accessibility of data, this method cannot study competition at the micro-level, leading to a limited understanding. Agent-based modeling (ABM) overcomes this limitation by simulating the actions and interactions of agents. From rule-based (Epstein and Axtell, 1996; Elliott and Kiel, 2002) to data-driven (Sajjad et al., 2016), and machine learning-based agents (Rand and Stummer, 2021), researchers dedicated to making agents appear more realistic. However, these agents cannot yet simulate complex human behavior, resulting in limitations to the authenticity of the simulation process.

Recently, the emergence of Large Language Models (LLMs) (OpenAI, 2023; Touvron et al., 2023; Zeng et al., 2023) provides an alternative to social simulations by enabling the creation of autonomous agents (Hardy et al., 2023; Jansen et al., 2023; Argyle et al., 2023; Ziems et al., 2023; Li et al., 2023b). An emerging body of work has explored these LLM-based agent approaches that simulated various society environments (Park et al., 2023; Gao et al., 2023; Törnberg et al., 2023; Liu et al., 2023; Akata et al., 2023), with primary focus on agents' *cooperation* and *collaboration* behaviors, such as software engineering and playing games (Wu et al., 2023; Xi et al., 2023; Abdelnabi et al., 2023). However, the work that examines the concept of *competition* is sparse. Han et al. (2023) studied firm competition and collusion, but only focused on price trend. To date, complex and realistic competitive simulation and study are still missing, which is important for a comprehensive understanding of the competition dynamics.

In this paper, we seek to address this research gap by investigating the *competition* between LLM-based agents. We

[1]University of Science and Technology of China [2]Microsoft Research [3]William & Mary [4]Georgia Institute of Technology [5]Carnegie Mellon University. Correspondence to: Jindong Wang <jindong.wang@microsoft.com>.

*Proceedings of the 41st International Conference on Machine Learning*, Vienna, Austria. PMLR 235, 2024. Copyright 2024 by the author(s).

first introduce a comprehensive framework for the study of agents' competition behaviors. This framework provides a structured and formal approach that is applicable to various scenarios. Guided by the framework, we develop a competitive practical environment (Figure 1) utilizing GPT-4 (OpenAI, 2023) to simulate a virtual town where two types of agents inhabit: restaurant and customers agents. Specifically, restaurant agents are responsible for managing restaurants and selling food to their customers. Customer agents play the role of judges by selecting restaurants and providing feedback on their experiences interacting with restaurants. Customers possess different characteristics, such as income, taste, health, and dietary restrictions, and are either an individual or a group. Within this simulated environment, restaurant agents compete with each other as they strive to attract and retain customers. This competition drives restaurant agents to evolve and adapt continuously and progressively. Restaurant agents develop innovative strategies to outperform their competitors.

We conduct micro- and macro-level analysis after running the simulation several times. Our key findings are:

- **Contextual Perception of LLM-based Agents**: We show that LLMs can accurately perceive competitive contexts and comprehensively analyze information, forming the basis for effective simulation experiments.

- **Market Strategy**: The behaviors observed in our environment conform to several classic sociological and economic theories, including *differentiation* (Porter, 1997), *imitation* (Lieberman and Asaba, 2006), *customer orientation* (Zeithaml et al., 2018), and *social learning* (Bandura and Walters, 1977).

- **Customer Decision**: We observe that customer decisions are usually influenced by several factors and vary from person to person, aligning with *consumer behavior* (Peter and Olson, 2010). Meanwhile, decision-making is different between individual and group dining.

- **Matthew Effect**: Our study reveals a Matthew Effect (Rigney, 2010) in the market competition, which manifests itself as a self-reinforcing cycle where popular restaurants gain even more popularity, while lesser-known restaurants continually receive less attention.

- **Customer Grouping Diminishes Winner-take-all**: We demonstrated that grouping customers can diminish "Winner-take-all" (Leadley et al., 2014) that is caused by Matthew Effect.

- **Competition improves product quality**: Our research demonstrates that competition among agents leads to improved product quality, which aligns with existing research (Lieberman and Asaba, 2006; Garvin, 1988).

The contributions of this paper are three-fold:

1. *A competitive framework for LLM-based agents.* We pioneered a comprehensive framework specifically designed to analyze competitive interactions between LLM-based agents.

2. *An implementation of a simulated competitive environment.* We developed a specialized competitive environment that allows structural and complex analysis of competition dynamics.

3. *Novel insights into competition dynamics.* We observed various competition behaviors from LLM-based agents that align with existing sociological and economic theories, informing future research and design implications.

## 2. Building the Competitive Environment

### 2.1. A general framework to study competition

Competition means that people need to compete for limited resources to make themselves thrive in an environment. We first propose a general framework for such study. As shown in Figure 2, our framework, referred to as "*CompeteAI*", consists of four major components.

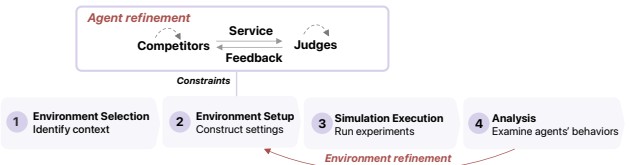

*Figure 2.* A general framework for studying the competition dynamics between AI agents. First, Choose an appropriate environment for LLM. Next, define each element, such as competitor and refinement method, in the environment to complete the setup. Finally, run the simulation and analyze results.

First, in environment selection, we identify an appropriate competition context for competition—this could range from competitive games, to company-customer interactions, and to other races as the main study environment. Second, in environment setup, we construct the chosen setting, leveraging the existing agent frameworks, such as CAMEL (Li et al., 2023a) or AutoGen (Wu et al., 2023) for adaptation. Third, in simulation execution, we run a series of experiments to capture the interaction processes between different agents within the established environment. Lastly, in analysis, we observe, analyze, and summarize the behaviors from the experimental results to derive insights.

Of note, the most important component is to create a competitive environment, where designers should meticulously consider the *competitors*, *judges*, and *interactions* between them (e.g., competitors provide *service* to the judges and judges provide *feedback* to the competitors). *Constraint* is necessary for this component to succeed, such as resource constraints and service constraints for the competitors, or

money constraints and buying constraints for the judges. The design of the constraints is inspired by the *resource dependence theory* (Hillman et al., 2009) where competition for resources can influence the behavior of an organization, relationships with other organizations, and strategies for survival and success. The design of these components highly depends on the competition situation. Designers should also pay attention to their interactions, iterations (since most competitions require feedback and rerun), and results management. Our framework serves as an ideal testbed for creating a diverse competitive environment to study the behaviors of agents. Detailed introduction of the components is in Appendix B.1.

## 2.2. Environment overview

Based on the framework, we implement the environment as a small town with two types of entities: 2 restaurants and 50 customers. Customers are either an individual or in a group (e.g., family, couple, or colleagues), detailed in Appendix C.2. We assume that each customer cannot cook and must go to one of the restaurants to eat. To simplify our observations, we assume that one customer should eat once in one restaurant every day. For profit, restaurants must compete to attract more customers. In this paper, both restaurants and customers are powered by LLM-based agents, which are GPT-4 (0613) (OpenAI, 2023). Specifically, each restaurant is managed by an agent to offer food to customers on a daily basis. The restaurant is operated via several pre-defined actions such as "modify menu", "manage chef", and "make advertisements" to serve customers of the day. Then, each customer receives the information from each restaurant and chooses between them. After their meal, customers leave comments as feedback to the restaurants. We set the simulation runs for 15 days, and if one of the restaurants decides to quit the race, the simulation will end.

There are three challenges in making this simulation practical. Firstly, for most LLM-based agents, their inputs and outputs are both in textual formats. It is non-trivial to enable them to interact with the real environment. Therefore, both restaurants and customers need real systems to emulate the possible actions. Secondly, agents should be sufficiently diverse to trigger more competitive behaviors. In the real world, users have diverse preferences. Some customers may prefer vegetarian food, while others prefer fast food. Thirdly, the validation is non-trivial. It is imperative to rigorously assess how well agents' behaviors within these simulations correspond to empirical human actions in real-world contexts. This ensures that the simulation is not only internally consistent but also externally valid.

In the following, we introduce how to overcome these challenges in our implementation.

## 2.3. Competitors

In this study, we employ agents as restaurant managers. Real-world restaurants involve complex operations like hiring staff, crafting menus, and advertising—tasks beyond the scope of text-based LLMs that lack real-world sensing capabilities (Dafoe et al., 2020). To address this, we use carefully designed prompts to contextualize the scenario for agents and build a comprehensive restaurant management system accessible with APIs (see Table 3 for details), which enables agents to manage the restaurant more effectively. For the ease of implementation and result analysis, we limit the competitive landscape to two restaurants. However, our framework can be readily used for more restaurants.

The process of a restaurant agent is described below: Each has a certain number of starting funds to hire chefs, make menus, make advertisements, and do other things. First, each agent receives recent daybooks recording the history of income, expenses, customer flow as well as comments on the last day. Information about its rivals (i.e., the other restaurant) from the last day is also provided, including the menu, customer flow, and comments. The agent then analyzes all information, designs, or revises strategy and planning for next day such as hiring a new chef or updating the menu. Then the agent interacts with the restaurant management system guided by the prompt to record the specified interaction method. After completing these operations, the agent summarizes them and stores this summary in memory for future planning. The main activities of the restaurant are shown in Appendix C.1.

## 2.4. Customers

Customers are judges in our environment, and it is important to include diverse customers to trigger more findings. To this end, we propose two variants: *characteristics* and *relationship* for each one. Characteristics comprise several factors: income, taste, health condition (e.g., diabetes), and dietary restrictions (e.g., vegetarians). All characteristics information is set by prompts and fed to the system to be stored as eternal characteristics. In terms of relationship, we set four common types: *family, colleague, couple,* and *friend*. Then, some customers are divided into groups that contain $2 \sim 4$ people according to their characteristics. Each person in a group is assigned a role (e.g., mother in a family) and the relationships with others are described. There are also differences between groups of the same type. For example, some family relations are harmonious while others are tense. In summary, we set 10 individual customers, 4 families, 4 colleagues, 3 couples, and 4 friends. Complete information of all customers is shown in Appendix C.2.

The process of each customer is as follows. Each day, information from two restaurants is shown to customers, including the name of the restaurant, customer score, adver-

tisement, menu, and comments. Each individual customer must choose one restaurant based on his/her characteristics, experience and the provided restaurants' information. For the group, the members first discuss where to go. During the discussion, each member can express their needs and ideas and then get a majority decision. In the decision phase, customers should provide reasons to better analyze their choices later. Then, the scores for the dishes saved in the restaurant system are sent to customers. Based on the scores of dishes and other information, each customer expresses the feeling that will become a dining experience. Some customers will leave comments including name, date, score, and content (in groups, all comments will be aggregated into one unified comment). Subsequently, these comments are stored and displayed to other customers.

### 2.5. Evaluating the quality of dishes

In our competitive environment, the quality of dishes plays a pivotal role in shaping the overall quality of the service. The quality of the dishes is associated with the price of the dish, the cost price, and the level of the chef. To gauge the quality of the dishes, we formulate several key assumptions to underpin our assessment: 1) The taste of the dishes exhibits a positive correlation with the skill levels of chefs, which is tied to their salary. 2) The quality and taste of the dishes are related to both the original and the selling prices.

Motivated by these assumptions, we introduce an empirical mechanism to evaluate the score $s$ for each dish: $s = 0.5 \times \frac{c}{p} + 0.5 \times \frac{f}{5000}$, where $c$ is the cost, $p$ is the price, and $f$ is the salary for the chef.

## 3. Results and Analysis

We run experiments 9 and 6 times for individual and group customers, respectively, due to the high cost of the simulation.[1] Our analysis consists of two perspectives: *micro-level* and *macro-level* analysis.[2]

Firstly, at the micro-level, we delve into the interaction between agents and simulated environment. Here, our focus is on assessing their fundamental capabilities in perception and action, and observing their behaviors. Secondly, at the macro-level, we examine the dynamic process and pay close attention to the system's evolution, identifying patterns within this evolution. We also analyze the outcomes of the simulation by evaluating the end results. In the two perspec-

---

[1]As a reference, the average API fee for running *single* once is $50, while the cost is $90 for *group* customers.

[2]Notably, some of the analyses below are illustrated through case studies for better interpretation. In fact, all analyses are based on data from all experiments, and the frequency of all observed behaviors is recorded in Table 2. For instance, imitation, a type of market strategies, occurred in all simulations.

tives, we not only align our observations with established theories from social sciences but also present interesting findings that offer promising avenues for further research.

### 3.1. Micro-level analysis: contextual perception

Perception enables the agent to continuously gather and interpret data, which is essential to understand the surrounding context, make informed decisions, and adapt to dynamic conditions. After observing how agents perceive and analyze the environment, we find that agents analyze the scenarios in a "shallow to deep" manner. For example, they sequentially analyze the trend of customer flow, dish feedback, and rivalry action. Then, they deeply analyze factors such as strategy effectiveness and market positioning.

We show a case study of a restaurant to support this finding:

```
Over the past few days, American Aroma has
displayed a growing trend in customer flow and
income, suggesting that our strategies are
resonating with the local clientele.  [...]
However, our dish scores have slightly fluctuated,
indicating room for improvement in the consistency
and complexity of flavors.  [...]  Our rival diner
has consistently good customer scores and comments,
particularly praising their BBQ Ribs Platter and
Fusion Bowl.  Their menu seems to strike a balance
between healthiness and hearty options, [...]
```

Through this example, we find that the agent is capable of analyzing observed information, verifying the correctness of strategy, and making adjustments accordingly. In conclusion, the agent effectively transitions from basic data analysis to a comprehensive evaluation of its performance and competitive standing, showcasing the ability to adapt and refine strategies based on a detailed understanding of customer preferences and market dynamics.

### 3.2. Micro-level analysis: market strategy

We then focus on the strategies taken by agents that are the critical element that determines which competitor can outperform others. We find that agents in our environment follow some classic market strategies including *differentiation*, *imitation*, *customer orientation*, and *social learning*.

**Differentiation.** Differentiation is a generic strategy that allows competitors to occupy a unique market position (Porter, 1997). Approaches to differentiation can take many forms: design brand image, customer service, or other dimensions. These approaches can also be observed in our environment. The following is a clip showing a competitor trying to focus on signature dishes to establish its own brand:

*Table 1.* Examples of customer needs and restaurant behaviors.

| Customer need | Agent behavior | Type |
|---|---|---|
| Vegetarian | Add "Vegan Delight Salad" to the menu | Dietary restrict |
| Diabetes | Add 'Sugar-free version Berry Parfait' | Dietary restrict |
| Seafood | Add "Grilled Seafood Platter" to the menu | Taste |
| Burger | Add "Classic American Burger" to the menu | Taste |
| Health Care | Introduce a "Local Favorites" section on the menu | Food Trends |

```
Streamline the menu to focus on a few high-quality,
signature dishes that can become customer favorites
and differentiate us from our competitor.
```

**Imitation.** Imitation is also a classic strategy that actively observes and adapts to the strategies of its competitors to maintain competitive parity or limit rivalry in market competition (Lieberman and Asaba, 2006). The following is another clip showing how another competitor finds its rival advantage and decides to imitate.

```
American Aroma 's emphasis on local ingredients and
healthful options is a clear advantage.   ...   Stars
& Stripes Diner will introduce locally sourced
ingredients for select dishes.
```

**Customer orientation.** Competitors discover and cater to customer needs to help them gain advantages in competition (Zeithaml et al., 2018). Those who prioritize customer insights are better positioned to adapt, innovate and thrive amidst competition. Table 1 shows the agent responses tailored to different customer needs. For instance, people with diabetes seek dishes with reduced sugar content, while seafood lovers prefer seafood dishes. Those needs exist in the comments, which are then received by the agents to make some arrangements to satisfy. Notably, competitors can not only identify individual customer needs, but also assess trends in customer factors (e.g., Health Care), allowing them to make adjustments accordingly.

### 3.3. Micro-level analysis: customer decision

The customer's decision plays a pivotal role in competition. In our analysis, the reasons behind customer preferences have been categorized and quantified, revealing that decisions are often influenced by a multitude of factors. This observation aligns with the theory of *consumer behavior* (Peter and Olson, 2010).

First, we summarized the reasons for different customers and categorized them into several primary topics. For example, dietary restrictions and taste preferences are grouped under "satisfying core needs". Choices based on high scores or positive reviews are classified as "considering the restaurant's reputation". Choices based on previous experiences

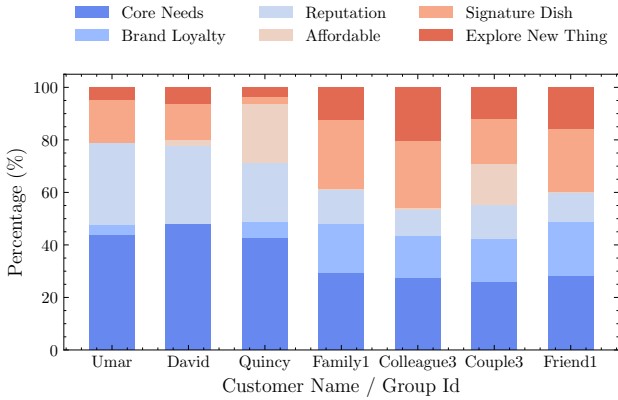

*Figure 3.* The distribution of reasons for customer decision. Customer considers multiple factors when making a decision, and conditions vary from person to person. In addition, groups are more inclined to explore new things while individual customers value reputation more.

are seen as "brand loyalty".

Based on this categorization, we counted the reasons behind the customer's decisions in all experiments. We randomly selected 3 single customers and 4 groups for presentation. Complete information is shown in Appendix C.2. As shown in Figure 3, it is evident that each individual customer or group considers multiple factors when making a decision, and conditions vary from person to person. In addition, a common factor is that "satisfaction of needs" weighs heavily on all customers. Furthermore, we can observe differences between individual customers and groups. For individual customers, the reputation of the restaurant is a crucial factor (avg 29.42) and ideas for exploring new things rarely appear (avg 7.18). In contrast, groups are more open to new dishes (avg 14.93) and they give less consideration to the restaurant's reputation (avg 10.71). The impact of these differences will be further discussed in 3.4.4.

### 3.4. Macro-level analysis

We present our macro-level analysis as follows: Strategy dynamics (§3.4.1), Matthew Effect (§3.4.2), Winner-take-all (§3.4.3), and product quality (§3.4.4).

#### 3.4.1. STRATEGY DYNAMICS

We have observed complex strategy dynamics, which refers to a series of dynamic interactions between companies striving for competitive advantages (Chen and Miller, 2012), emerging in the competition. These dynamics are driven by an interplay of differentiation and imitation behaviors.

*Overall Findings*: As shown in Figure 4, on Day 2, R1 first proposes the use of local ingredients in dishes to appeal to health-conscious customers. During the next two days, the selling point helps R1 attract a large number of customers.

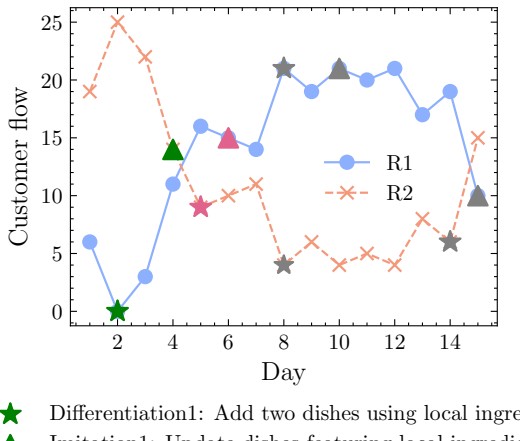

★ Differentiation1: Add two dishes using local ingredients
▲ Imitation1: Update dishes featuring local ingredients
★ Differentiation2: Add 'Stars & Stripes Fusion Bowl'
▲ Imitation2: Add 'American Fusion Bowl'
★ Other Differentiations
▲ Other Imitations

*Figure 4.* A Case Study of Competitive Dynamics.Imitation and Differentiation among restaurants creates a competitive dynamic that ultimately maintains a dynamic equilibrium.

Realizing the great success of these selling points, R2 updates some of its dishes featuring local ingredients on Day 4 and further introduces 'Stars & Stripes Fusion Bowl' to support customized services for customers on Day 5. Subsequently, R1 adds 'American Fusion Bowl' to benchmark against R2. After this, two agents continue to look for new selling points to create differentiation while imitating the good selling points of their rivals.

*Core Manifestation*: Competitors often rely on differentiation to win advantages. However, the risk is that it can be easily imitated by competitors, reducing differentiation (Porter, 1997). Therefore, the advantage can usually only be maintained for a limited period, and the competitor needs to continually differentiate to gain a competitive edge.

*Dynamic equilibrium*: If two restaurants share the same settings (cuisine type, initial funding), their menus naturally tend to be similar. However, for differentiation, competitors have been introducing new elements in menus that reduce the similarity between menus while their rivals' imitation increases it, ultimately leading to a dynamic equilibrium. As shown Figure 5(a), we calculated the similarity between the menus of the two restaurants for each day during all the experiments and then averaged the similarity for each day. We found that the similarity of the menus remained constant around 36%.

### 3.4.2. MATTHEW EFFECT

We observed a phenomenon reminiscent of the Matthew Effect (Rigney, 2010), wherein entities with an initial competitive edge continue to accrue benefits, leaving others in a

perpetual state of catch-up, leading to unequal growth and opportunities. This effect is widely recognized in various domains, including education (Walberg and Tsai, 1983) and science funding (Bol et al., 2018). Below, we elaborate on how our findings offer practical insights into the manifestation of the Matthew Effect in the context of LLM-based agents, specifically within the dynamics of restaurant customer traffic and feedback mechanisms.

*Overall Findings:* As shown in Figure 5(b), on Day 1, the majority of customers choose R1 due to its affordability, diverse menu offerings and other factors, and the high quality of the R1 dishes gives them a satisfying experience. As a result, R1 receives positive customer comments and high customer score (average 7.2). In contrast, R2 has few customers, which means fewer comments. What is worse is that customer comments are mixed and customer scores (average 6.0) are lower than R1 due to the quality of the dishes. On Day 2, for R1, higher scores, more positive comments, and a revised menu attract new customers and encourage existing customers to stay. This pattern persists daily, exacerbating R2's situation.

*Core Manifestation:* R1's initial success reinforces its advantage through a positive feedback loop: more comments allow R1 to obtain more feedback, enabling better adjustments. Additionally, higher customer scores and more positive comments help R1 establish a good reputation among customers. This dual helps R1 attract more customers. On the contrary, due to fewer customers, R2 receives limited feedback. Additionally, any adjustments made by R2 might not produce immediate noticeable results due to the small customer base. R2 struggles to break this cycle, highlighting the disparity in growth and success.

*Disproportionate Growth Patterns:* The evolving dynamics, where R1 thrives and R2 faces challenges, epitomize the uneven growth trajectories central to the Matthew Effect.

In short, our findings underscore the profound impact of initial advantages and the pivotal role of feedback in creating a self-perpetuating cycle of success for some and challenges for others, aligning with the Matthew Effect.

### 3.4.3. CUSTOMER GROUPING DIMINISHES WINNER-TAKE-ALL

The "Winner-take-all" phenomenon (Leadley et al., 2014) occurs due to the Matthew effect. We define the winner-take-all as follows. After five days of competition, one restaurant has more than 80% of the customers until the competition ends (Day 15). By conducting a statistical analysis of this phenomenon, we observe that the winner-take-all happens more frequently in *single* customers (66.7%) and rarely for *group* customers (only happened once, which is 16.7%). We conclude that the phenomenon is due to one of the

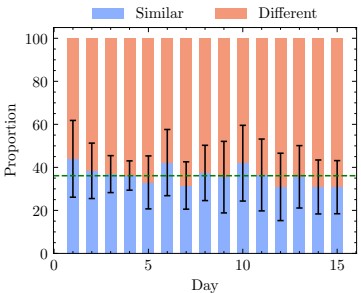

(a) Dynamic Equilibrium in Menu Similarity Between Two restaurant Over a 15-Day Period. The similarity remain constant around 36%.

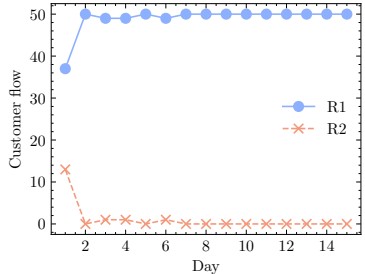

(b) A Case Study of The Matthew Effect. With an initial competitive edge, R1 has held almost all of the market.

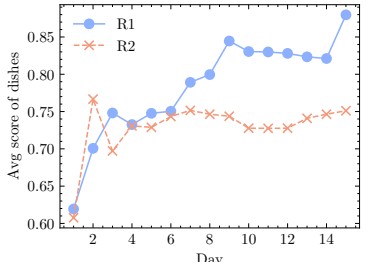

(c) The Trend of Dishes Score. From Day 1 to Day 15, the average increase in dish scores is 0.26 for R1 and 0.22 for R2.

*Figure 5.* Aggregated results during the simulation experiment.

results shown in Figure 3, which shows that *groups are more inclined to explore new things and don't consider the reputation as a key indicator* .

```
I totally get the appeal of trying something new,
and American Aroma does have a few dishes that
catch my eye.
```

The preference of groups gives disadvantaged restaurants a chance to get their dishes noticed, implement effective strategies, and gather feedback for improvement. These experimental customers may also recommend the restaurant to others through their comments. This disrupts the previously established positive feedback mechanism of the Matthew effect, thus diminishing winner-take-all.

### 3.4.4. COMPETITION IMPROVES THE PRODUCT QUALITY

An interesting phenomenon is that, in competition, the quality of the restaurant's food usually gets better and better. This phenomenon is aligned well with related research (Lieberman and Asaba, 2006; Garvin, 1988).

We show the improvement in quality through two aspects: first, the frequency with which the average score of dishes in at least one restaurant improves with time in the competition is 86.67%, indicating that, with high probability, customers are likely to have a better dining experience in one of the restaurants compared to before. Then, Figure 5(c) also supports this result: the average score of the dishes increased with time. From Day 1 to Day 15, the average increase in dish scores is 0.26 for R1 and 0.22 for R2.

We find that competition is the key factor in this improvement. In a highly competitive market, customers have more options, forcing competitors to focus more on improving service quality. At the same time, due to the presence of rivals, competitors must strive to raise their standards to gain a competitive edge. This dynamic environment ultimately drives competitors to improve the quality of dishes.

*Table 2.* From the observed phenomena to theories and the occurrence frequency in experiments.

| Phenomenon | Theory | Frequency |
|---|---|---|
| Differentiation | Market Competition | 100% |
| Imitation | Market Competition | 100% |
| Customer orientation | Market Competition | 100% |
| Strategy dynamics | Market Competition | 100% |
| Product quality improvement | Market Competition | 86.67% |
| Matthew Effect | Sociological Theory on Matthew Effect | 66.7% (single), 16.7% (group) |

Next, a piece of history record:

```
To enhance dish flavors, incrementally increase
the original prices of popular dishes to source
even higher-quality ingredients while keeping cost
ratios reasonable for customer satisfaction.
```

## 4. Discussion

**Alignment with existing theories and why?** As shown in Table 2, a series of observed phenomena align well with existing sociological and market theories. Phenomena at micro-level (Differentiation, Imitation, Customer orientation) are manifestations of agent endogenous behaviors. But *why* agents possess these behaviors are unexplored due to the black-box nature of the large language models we adopted (GPT-4). A possible explanation could be that the models are well trained on a massive corpus that contains texts from various disciplines such as psychology, sociology, and economics (OpenAI, 2023). Therefore, we doubt that the model could have already memorized these popular theories and examples, leading to these "common" behaviors triggered by our prompts.

**Beyond the alignment.** An interesting question is: can LLM-based agents behave *more than* just following existing knowledge in the training data? Can they cultivate *new* intelligence? We believe that this could be profoundly impor-

tant in performing new studies in sociology and economics, leveraging agents to uncover new rules, laws, or even theories. Furthermore, observed behaviors are well aligned with existing theories, indicating that they are also aligned with human values (Gabriel and Ghazavi, 2021), which may trigger interest from the value alignment community to conduct research in an agent-based environment. This work can then be seen as the baseline for such alignment research, and more complex algorithms can be introduced.

**Broader implications for AI adoption.** Recognizing the presence of Matthew Effect in LLM competition can inform strategies for adopting and improving newer or smaller LLM agents. By understanding the challenges they might face due to initial disadvantages, strategies can be developed to level the playing field. The Matthew Effect, when observed in the realm of LLMs, can lead to monopolistic behaviors or concentrated power among a few dominant models. Recognizing this effect is crucial for ensuring diversity, fairness, and broad access in the AI landscape. By understanding the dynamics of the Matthew Effect in LLM-based competition, researchers and developers can better design training protocols, feedback mechanisms, and integration strategies to ensure that even agents with initial disadvantages have the opportunity to thrive.

## 5. Related Work

**Empirical studies on competition.** The method, through research on competitive phenomena in the real world, have revealed several patterns and rules, providing valuable insights into the dynamics of competition (Porter, 2008; Kosfeld and Von Siemens, 2011). For instance, Markussen et al. (2014) found that inter-team competition can serve as a catalyst for intra-team cooperation by stimulating improvements in relative group performance. Chen (2008) further highlighted the intricate interplay between cooperation and competition in real-world scenarios. Rigney (2010) proposed the "Matthew Effect" which revealed competitive phenomena in academia. The effect indicates that well-known scholars are more likely to receive resources, honors, and citations, while new scholars face greater competitive pressure. These studies are based on observations and analysis of real-world situations and cannot independently control variables. Additionally, collecting comprehensive data is challenging, resulting in some important phenomena inadequately studied.

**Large Language Model-empowered Agent-based Modeling** Due to the powerful capabilities and human-like behaviors exhibited by large language models, numerous researchers have begun applying LLM-based agents within Agent-based Modeling (ABM) to construct more intelligent agents and more realistic, intricate simulation scenarios. As a pioneering work, Generative Agent (Park et al., 2023) established a village composed of 25 agents. This work

systematically designed the agent architecture within the simulation environment, setting a foundational framework for future agent designs. Additionally, this study explored the phenomena and mechanisms of information dissemination in the simulation, marking a significant milestone in applying LLM-based agents to ABM. Wang et al. (2024) developed a virtual recommendation system environment to investigate phenomena such as filter bubbles and user conformity. Li et al. (2024) applied LLM-based agents to a macroeconomic environment, successfully replicating real-world phenomena that traditional simulation methods have struggled to reproduce.

Significant advancements have also been made in the area of collaborative cooperation. CAMEL (Li et al., 2023a) proposed a framework for agent cooperation featuring a commander for planning and executors for task implementation. Qian et al. (2023) created a virtual software company where agents assumed roles such as CEO and engineer, collaborating to complete software development projects. Zhang et al. (2023) delved into the cooperation mechanisms among agents, providing insights from a social psychology perspective.

Despite the progress in cooperative mechanisms, research on competition mechanisms remains limited. Chen et al. (2023) constructed an auction scenario to evaluate the competitive planning and execution abilities of LLMs, but the study focused more on these capabilities than on analyzing the behaviors exhibited by LLMs or the dynamic changes within the system. Han et al. (2023) examined corporate competition and cooperation, concentrating primarily on price dynamics. These studies fall short of simulating complex competitive environments and thoroughly exploring competitive behaviors and system evolution. Our research aims to fill this critical gap.

## 6. Limitations and Future Directions

While this study offers a valuable initial exploration of LLM-based agents in a competition scenario, it should be considered a stepping stone for more comprehensive research in this domain. (1) Sample Size and Diversity. Due to the constraints imposed by the GPT-4 API limitations, our experiments did not involve a significant number of restaurants and customers. (2) Text-Based Interactions. Our current framework leverages GPT-4, the most adept text-based Language Learning Model (LLM), which predominantly relies on textual data. We acknowledge that real-world environments often involve multi-modal interactions and inputs, such as image, video, and audio. As more sophisticated multi-modal LLMs become publicly available on a large scale, we anticipate that future studies could offer a more holistic view. (3) Version-Specific Findings. The results in this paper are based on GPT-4-0613. We acknowledge that

future API updates may affect the results.

## 7. Conclusion

We introduced a general framework, CompeteAI, to study the dynamics of competition using LLM-based agents. By instantiating the framework as a virtual town with restaurant and customer agents, we extensively explored the competition behaviors of agents. Our study revealed several interesting findings in accordance with classic sociological and economic theories. To conclude, our work confirmed that LLM-based agents can be used to simulate a competitive environment, providing research experience for future studies on sociology, economics and human studies.

## Impact Statement

We leveraged LLM-based agents to generate plans for running a restaurant or writing a comment. Our study does not output any irresponsible or risky words.

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

# A. Preliminaries

**Social learning theory** (Bandura and Walters, 1977) posits that individuals learn new behaviors through observation, imitation, and modeling. This paradigm has found applications in various domains such as psychology, education, and sociology (Latham and Saari, 1979; Deaton, 2015; Davis and Luthans, 1980). It serves as a robust framework for comprehending the intricate interplay between individual cognition and external influences in learning.

In this work, we explore the application of social learning theory by scrutinizing the behaviors of LLM-based agents in an interactive environment. Through comprehensive experiments, we successfully elucidate how LLM-based agents exhibit efficient social learning behaviors, and establish their potential utility in simulating complex dynamics in various disciplines social science.

**Market competition theory** (Smith, 1937) elaborates on how companies and organizations compete for consumer attention and finite resources within a marketplace (Smith, 1937), which plays a vital role in understanding economic dynamics, shaping business strategies, and informing public policy decisions (Hirshleifer, 1978).

In this study, we delve into the applicability of market competition theory by investigating how competition among LLM-based agents influences their learning processes and decision-making mechanisms. Through meticulous empirical study, we have found compelling evidence that when these agents engage in competitive environments, they exhibit substantial enhancements in service quality and capacity to adapt their strategies to meet the diverse and evolving needs of their customers. These findings underscore the importance of embracing customer-centric strategies to achieve success in competitive market landscapes.

# B. Environment

## B.1. Details of the framework

In this section, we introduce the key concepts of our framework, including environment, competitors, judges, constraints, service and feedback, and agent refinement.

- **Environment:** The simulated space where competitions occur, typically facilitated by LLM-based agents.
- **Competitors:** The primary subjects who perform certain actions to gain advantages, such as attracting more customers or securing more votes.
- **Judges:** Entities that receive services from competitors and influence their success, such as customers in a retail setting or voters in an election.
- **Constraints:** Rules designed to level the playing field in competitions. Examples include limiting dining choices to one restaurant per meal or one vote per person in elections.
- **Service and Feedback:** Competitors offer services to win over judges, who in turn provide feedback that informs future competitor actions.
- **Agent Refinement:** Both competitors and judges adapt based on interactions, such as updating strategies or sharing information among peers.
- **Environment Refinement:** The design of the environment could further be refined according to the process of the study to better simulate the real-world scenarios and achieve the trade-off between simulation resources (API fees, hardware and software constraints) and real-world scenarios.

## B.2. Our environment

The small virtual town of our environment is shown in Figure 6.

# C. Implementation of Restaurant and Customer Agents

## C.1. Restaurant agent

The flow of the restaurant agent is shown in Figure 7, and Table 3 shows the actionable API.

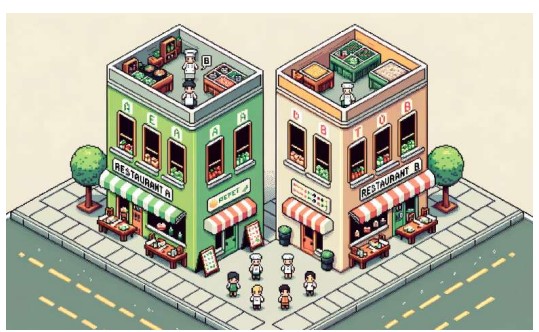

⮞ **Environment:** A virtual town with only restaurants and customers.
⮞ **Competitors:** Two LLM-based agents (i.e., restaurants).
⮞ **Judges:** LLM-based agents as customers with different characteristics.
⮞ **Constraints:**
   1) Customers cannot cook
   2) One customer should have one meal every day at one restaurant
   3) There are housing and facilities costs for the restaurants every day
   4) There will only be the starting fund for each restaurant to operate
⮞ **Service & Feedback:** Restaurants provide food to the customers and customers leave comments to the restaurants.
⮞ **Agents Refinement:** Restaurants update their menu or operations based on customer feedback. Customers update their comments to restaurants.

*Figure 6.* Our simulated virtual town consists of two types of agents: restaurant agents that are served as competitors and customer agents as examples of judge agents. Key concepts of our proposed framework include Environment, Competitors, Judges, Constraints, Service and Feedback, and Agents Refinement. Environment refinement is not included here since this is out of the scope of the virtual town.

*Table 3.* The action space (APIs) that agents can leverage.

| API | Properties | Action Space |
|---|---|---|
| basic_info | name, rent, money, status | Get information & Modify restaurant name |
| chef | name, salary | Hire / Fire chef & Adjust chef salary |
| menu | name, price, cost_price, description | Add / Delete / Get / Modify item in menu |
| advertisement | content | Get / Modify advertisement |
| comment | day, name, score, content | Get all comments |
| daybook | profit, expense, num_of_customer, and so on | Get daybooks |

**RULES**

Please keep in mind the following **rules**:
1. Chefs
— You are not allowed to communicate with or train the chefs
...

2. ...

**BASIC INFORMATION PROMPT**

Now your restaurant basic information as below:
< Current Basic Information >

If you want to modify the restaurant name, your response must follow the following format:
< Designed JSON format> ...

**DAYBOOK & RIVAL INFO PROMPT**

Today is day <x>, the daybook is as below:
< Profits, Expenses, Number of customers, ... >

The customer comment as below:
< Designed JSON format> ...

The rival restaurant info as below:
< Number of customers & menu of rival ...>

START

Ideation

*A day later...*

Restaurant agents

Action List

Basic Info

Menu

Chef

Advertisement

**MENU PROMPT**

Now your **restaurant basic information** as below:
< Current Basic Information >

If you want to modify the restaurant name, your response must follow the following format:
< Designed JSON format> ...

**ADS PROMPT**

Welcome to the **Advertisement** Management Module. Now your Advertisement as below:
< Current Ads >

You can perform the following actions:
1. Update this Advertisement:
< Designed JSON format> ...

**CHEF PROMPT**

Welcome to the **Chef** Management Module. Now your chef info as below
< Current Basic Information >

You can perform the following actions:
1. Hire a new chef
< Designed JSON format> ...

*Figure 7.* An overview of the process of operating restaurants among the competitors (i.e., two restaurants). On each day, the restaurant receives the daybook and the information for the rival. Then, the agent manages the restaurant prompted by the basic information prompt, menu prompt, chef prompt, and ads prompt. More details are in the main text.

## C.2. Customer agent

The flow of the customer agent is shown in Figure 8.

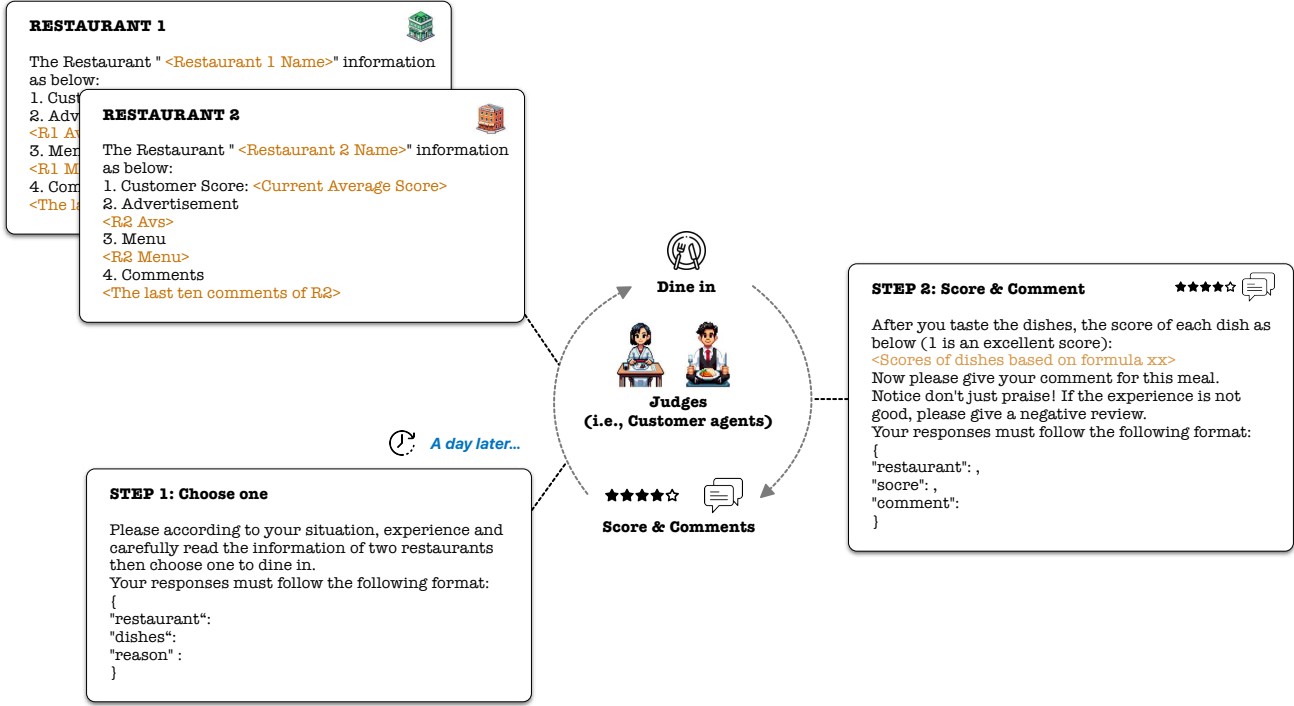

*Figure 8.* The detailed customer dining process. On each day, each customer receives the information from two restaurants and selects one to dine. After meal, customers leave comments and scores.

The details of all customers are shown in Table 4 and the details of all groups are shown in Table 5.

## D. Detailed results

The distribution of restaurant selection for all customers is shown in Figure 9.

The average distribution of reason for *single* and *group* is shown in Table 6.

*Table 4.* Detailed information of the customers.

| Name | Income | Taste | Health | Dietary Restriction | Personality |
|------|--------|-------|--------|---------------------|-------------|
| Alice | $12000 (Affluent) | Local comfort foods | Healthy | None | Easy-going |
| Amelia | $8700 (Middle Class) | Mexican food | Healthy | None | Spirited |
| Bob | $8000 (Middle Class) | Rice and noodle dishes | No concerns | None | Strict |
| Brian | $6200 (Poor) | Street food | Healthy | None | Resourceful |
| Charlie | $15000 (Affluent) | Sandwiches and salads | Healthy | None | Picky |
| Chloe | $10300 (Middle Class) | Indian cuisine | Diabetic | Low sugar | Thoughtful |
| David | $10000 (Middle Class) | Breakfast foods | High blood pressure | Low sodium | Cheerful |
| Dexter | $13800 (Affluent) | Barbecue | Healthy | None | Sociable |
| Emma | $5800 (Very Poor) | Organic food | Healthy | None | Optimistic |
| Eve | $7000 (Poor) | Simple dishes | Healthy | None | Shy |
| Felix | $11700 (Middle Class) | Chinese cuisine | Healthy | None | Analytical |
| Frank | $9500 (Middle Class) | Fast food | Healthy | None | Adventurous |
| Giselle | $9400 (Middle Class) | Desserts | Healthy | None | Creative |
| Grace | $11000 (Middle Class) | Soups and stews | Diabetic | Low sugar | Friendly |
| Henry | $14000 (Affluent) | Meat | Healthy | None | Reserved |
| Hugo | $14200 (Affluent) | Gourmet burgers | Healthy | None | Leader |
| Iris | $7600 (Poor) | Salads | Healthy | None | Gentle |
| Ivy | $6500 (Poor) | Seafood | Healthy | None | Outspoken |
| Jack | $8500 (Middle Class) | Steak and meat dishes | Healthy | None | Energetic |
| Jake | $6800 (Poor) | Fried food | High blood pressure | Low sodium | Jovial |
| Katie | $5000 (Very Poor) | Vegan dishes | Healthy | None | Compassionate |
| Lara | $8900 (Middle Class) | Seafood | Healthy | None | Ambitious |
| Leo | $13500 (Affluent) | Pasta and pizza | Healthy | None | Relaxed |
| Maggie | $9000 (Middle Class) | Chocolate and sweets | Healthy | None | Carefree |
| Max | $5300 (Very Poor) | Plant-based meals | Healthy | None | Resourceful |
| Nate | $7500 (Poor) | Grilled dishes | Healthy | None | Meticulous |
| Nora | $12800 (Affluent) | Fine dining | Gluten intolerance | Gluten-free | Elegant |
| Olivia | $13000 (Affluent) | Mediterranean cuisine | Allergies | Gluten-free | Artistic |
| Oscar | $9600 (Middle Class) | Traditional cuisine | Healthy | None | Reserved |
| Paula | $11200 (Middle Class) | Greek food | Healthy | None | Outgoing |
| Peter | $6000 (Poor) | Baked goods | Healthy | None | Curious |
| Quincy | $6400 (Poor) | Fast food | Overweight | Low calorie | Easygoing |
| Quinn | $8200 (Middle Class) | Spicy food | Healthy | None | Bold |
| Rachel | $14500 (Affluent) | Gourmet dishes | Lactose intolerant | Dairy-free | Sophisticated |
| Ruby | $14800 (Affluent) | Sushi | Healthy | None | Discerning |
| Sam | $5500 (Very Poor) | Home cooking | Healthy | None | Warm |
| Steve | $7900 (Poor) | Comfort food | High cholesterol | Low fat | Friendly |
| Tara | $10500 (Middle Class) | Exotic fruits | Healthy | None | Adventurous |
| Tina | $10800 (Middle Class) | Mediterranean cuisine | Healthy | None | Charismatic |
| Ulysses | $13300 (Affluent) | International cuisine | Healthy | None | Explorer |
| Umar | $12500 (Affluent) | Grilled seafood | High cholesterol | Low cholesterol | Discerning |
| Valerie | $8300 (Middle Class) | Organic foods | Healthy | None | Intellectual |
| Vicky | $7300 (Poor) | Comfort food | Healthy | None | Easygoing |
| Wade | $5700 (Very Poor) | Simple meals | Healthy | None | Hardworking |
| William | $11500 (Middle Class) | Sushi and Japanese cuisine | Healthy | None | Reserved |
| Xavier | $11900 (Middle Class) | Caribbean cuisine | Healthy | None | Vibrant |
| Xena | $9800 (Middle Class) | Italian cuisine | Healthy | None | Lively |
| Yara | $9200 (Middle Class) | Vegetarian dishes | Vegan | Vegan | Compassionate |
| Yasmine | $7800 (Poor) | Vegan options | Healthy | Vegan | Compassionate |
| Zach | $12200 (Affluent) | French cuisine | Gluten sensitivity | Gluten-free | Connoisseur |

Table 5. Detailed information of the customer groups.

| Type | Feature | Members |
|---|---|---|
| Family | Affluent, Harmonious | Rachel(Mother), Henry(Father), Ruby(Daughter), Hugo(Son) |
| Family | Strained, Tense | William(Father), Paula(Mother), Felix(Eldest Son), Xavier(Younger Son) |
| Family | Poor, Strained | Nate(Father), Vicky(Mother), Steve(Son) |
| Family | Very Poor, Close-knit | Wade(Father), Ivy(Mother), Emma(Daughter) |
| Colleague | High-profile Job, Peer | Dexter, Ulysses |
| Colleague | Financially Constrained, Superior-Subordinate | Yasmine (Supervisor), Subordinate (Eve) |
| Colleague | Middle-Class, Peer Competition | Chloe(Leadership) , Tara(Subordinate), Tina(Peer) |
| Colleague | Middle-Class, Peer Collaboration | Frank, Giselle, Yara |
| Couple | Affluent, Romantic | Nora (Girlfriend), Alice(Boyfriend) |
| Couple | Navigating Challenges | Maggie(Girlfriend), Valerie(Boyfriend) |
| Couple | Long-term, Financial Struggles Strong Bond | Max(Boyfriend), Sam(Girlfriend) |
| Friend | College Days | Olivia, Charlie |
| Friend | Middle-class, Childhood Friends | Grace, Peter |
| Friend | Middle-class, High School Friends | Amelia, Lara |
| Friend | Childhood Friends, Different Background | Jake, Brian, Quinn |

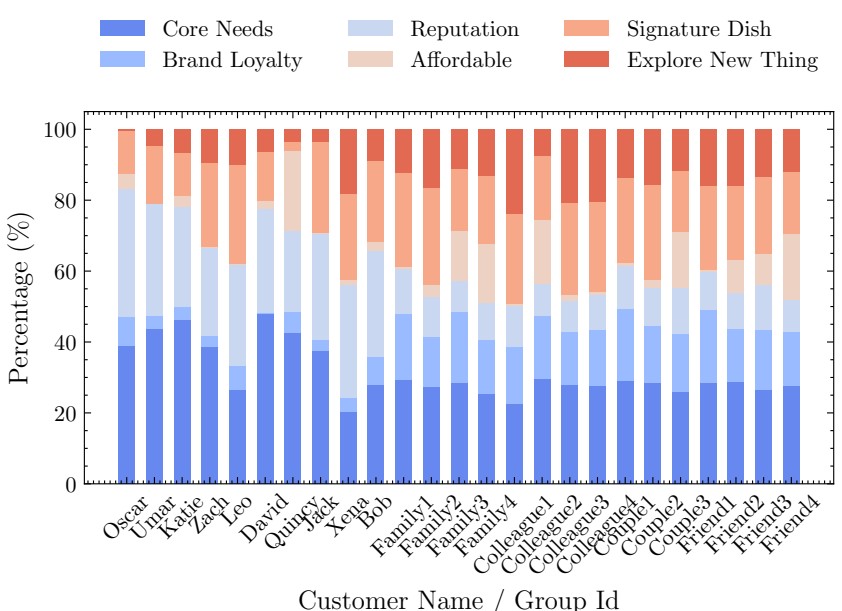

Figure 9. The distribution of reasons for customer decision.

Table 6. Detailed information of the customers.

| Type | Core Needs | Brand Loyalty | Reputation | Affordable | Signature Dish | Explore New Thing |
|---|---|---|---|---|---|---|
| Individual | 37.03 | 4.63 | 29.42 | 3.60 | 18.14 | 7.18 |
| Group | 27.56 | 16.89 | 10.71 | 7.48 | 22.43 | 14.93 |

