# OpenReview forum: "CompeteAI: Understanding the Competition Dynamics of Large Language Model-based Agents"
_ICML.cc/2024/Workshop/Agentic_Markets — Agentic Markets @ ICML'24 Poster_

### Official Review · Reviewer_bUEB · 2024-06-15
**This is an interesting simulation exercise using LLMs to understand competition. The paper is ambitious but the exercise is neither comprehensive now very insightful.**

**Rating:** 3
**Confidence:** 4

**Review:**

This paper simulate a market where restaurants compete. LLMs are used to create restaurant and consumer agents. This is a valid simulation exercise, but the results are neither conclusive nor insightful. It is unclear what we take away other than some descriptions based on a limited-scale simulation. It is also unclear what the difference would be had one used other non-LLM-agent-based simulations. Given the high standard of the conference, I am afraid the paper falls short.

---

### Official Review · Reviewer_4JuV · 2024-06-17
**Interesting paper**

**Rating:** 8
**Confidence:** 4

**Review:**

Summary: The paper introduces simulated competition among agents in a situation that closely follows the real world example. Through the experiment conducted in the paper, they successfully extracted interesting behaviors and final outcomes of competition which resembles that of real world.

Pros:

Using LLMs to closely mimic the real world, making almost the best effort to encode real world situation such as character and relationships into prompts. The simulation outcome conforms with the real world outcome of competition.

The split of micro level and macro level analysis shows a clear behavior of decisions made by agents

Cons:

Lacks reasons for choosing character and relationships for diversifying customer agents.

Empirical mechanism to evaluate score for dishes needs some backup data for empirical decision.

Reasoning for next decision taken by agents should be analyzed in more detail.




Suggestions:

Objective reasoning for choosing character and relationships of customer agents

Explain empirical mechanism

Analyze the importance of micro vs macro level feedback in taking next strategy.
Possibility of collusion between restaurants should be addressed or discussed. If not present in the simulation, state how it was prevented.

---

### Official Review · Reviewer_Gska · 2024-06-18
**CompeteAI: Understanding the Competition Dynamics of Large Language Model-based Agents**

**Rating:** 8
**Confidence:** 4

**Review:**

The paper explores the competitive dynamics among large language model (LLM)-based agents. The authors present a framework and a simulated environment to study these dynamics, specifically focusing on competition rather than cooperation. The simulation involves restaurant agents competing to attract customer agents within a virtual town. Key findings align with classic sociological and economic theories, providing insights into competitive behaviors, market strategies, and customer decision-making.

Strengths
Innovative Framework: The proposed framework for studying competition among LLM-based agents is novel and well-structured, offering a new perspective in agent-based modeling.
Interesting Simulation Environment: The virtual town setup with restaurant and customer agents effectively simulates real-world competitive scenarios, allowing for detailed analysis of agent behaviors.
Alignment with Theories: The findings align with established sociological and economic theories, adding credibility and depth to the study.
Comprehensive Analysis: Both micro- and macro-level analyses provide a thorough understanding of the agents' behaviors and the dynamics of the competitive environment.
Detailed Methodology: The paper provides a clear and detailed description of the simulation setup, agent behaviors, and the analysis process.
Weaknesses
Limited Scope: The study is limited to a specific type of competition (restaurant management) and a small number of agents, which may not generalize well to other competitive scenarios.
Sample Size: The small number of agents (two restaurants and 50 customers) limits the robustness of the findings and may not capture the full complexity of competitive dynamics.
Questions
Generality of the Framework: How adaptable is the proposed framework to other types of competitive scenarios beyond restaurant management?
Scalability: How would the findings change with a larger number of agents and a more complex competitive environment?
Recommendation
Rating: 8: Top 50% of accepted papers, clear accept
Confidence: 4: The reviewer is confident but not absolutely certain that the evaluation is correct